# Cepharanthine Enhances MHC-I Antigen Presentation and Anti-Tumor Immunity in Melanoma via Autophagy Inhibition

**DOI:** 10.3390/cells14161231

**Published:** 2025-08-09

**Authors:** He Luo, Dan Chen, Jing Zhou, Dingye Wang, Qingsong Du, Qianwei Cai, Sixian Lv, Xu Zhao, Guangxian Zhang, Yuhui Tan, He Jin, Xiaoyi Liu, Hua Yi, Jieying Guan

**Affiliations:** 1Research Center of Integrative Medicine, School of Basic Medical Sciences, Guangzhou University of Chinese Medicine, Guangzhou 510006, China; luohe@stu.gzucm.edu.cn (H.L.); chendan@stu.gzucm.edu.cn (D.C.); zhoujing@oncolab.cn (J.Z.); wangdingye@oncolab.cn (D.W.); duqingsong@oncolab.cn (Q.D.); caiqianwei@oncolab.cn (Q.C.); lvsixian@oncolab.cn (S.L.); zhaoxu@oncolab.cn (X.Z.); zhangguangxian@gzucm.edu.cn (G.Z.); tyuhui@gzucm.edu.cn (Y.T.); 021103@gzucm.edu.cn (H.J.); liuxiaoyi@gzucm.edu.cn (X.L.); 2Department of Pathology and Pathophysiology, Guangzhou University of Chinese Medicine, Guangzhou 510006, China; 3Department of Biochemistry, Guangzhou University of Chinese Medicine, Guangzhou 510006, China

**Keywords:** MHC-I, antigen presentation, tumor immune escape, cepharanthine, autophagy

## Abstract

Major histocompatibility complex class I (MHC-I)-mediated antigen presentation plays a pivotal role in anti-tumor immunity by enabling CD8^+^ T cells to recognize and eliminate malignant cells. In melanoma, modulation of this pathway is critical for improving the efficacy of immunotherapies. Our study demonstrates that the natural compound Cepharanthine (CEP) exhibits notable antitumor activity by enhancing MHC-I-mediated antigen presentation. CEP treatment upregulated MHC-I expression (both membrane-bound and total levels) in melanoma cells in a concentration-dependent manner, thereby improving antigen-presenting capacity. Interestingly, when autophagy was pharmacologically blocked using Bafilomycin A1, co-treatment with CEP did not lead to further elevation of MHC-I expression, suggesting that CEP’s effect is mediated through disruption of the autophagic pathway. Mechanistically, CEP induced autophagosome accumulation, as evidenced by an increase in GFP-LC3 puncta. Fluorescence imaging further confirmed that CEP selectively impaired lysosomal acidification without affecting autophagosome–lysosome fusion, thereby inhibiting late-stage autophagic flux. Furthermore, CEP treatment promoted CD8^+^ T cell infiltration into tumor tissues and enhanced the antitumor efficacy of anti-PD-1 therapy, resulting in greater tumor suppression compared to either treatment alone. The study elucidates how CEP’s selective lysosomal inhibition creates a tumor microenvironment more susceptible to immune surveillance, primarily through preserved MHC-I surface expression and subsequent T cell recognition. This work highlights CEP as a promising immunomodulatory agent and provides a potential strategy for improving the outcomes of immune checkpoint blockade therapy.

## 1. Introduction

Melanoma is a malignant skin tumor arising from melanocytes, characterized by high invasiveness and poor prognosis. Global epidemiological data indicate a rising incidence of melanoma, with an increasingly younger age of onset, while the 5-year survival rate remains dismally low [1]. Current standard treatments—including surgical resection, chemotherapy, radiotherapy, and cytokine therapy—offer limited efficacy in advanced-stage patients, highlighting the urgent need for novel therapeutic approaches.

In recent years, the advent of cancer immunotherapy has revolutionized melanoma treatment, significantly improving clinical outcomes [2,3]. As a key strategy in modern oncology, immunotherapy primarily functions by activating or enhancing the immune system’s capacity for tumor-specific recognition and cytotoxicity [4]. Among immune effector mechanisms, activated CD8^+^ T cells mediate tumor cell killing by recognizing antigenic peptides presented by major histocompatibility complex class I (MHC-I) molecules on the surface of tumor cells. This underscores the central role of MHC-I-mediated antigen presentation in anti-tumor immunity. However, the immunosuppressive tumor microenvironment often impairs this process. Previous studies have shown that melanoma cells frequently downregulate MHC-I expression, with some cases exhibiting complete MHC-I loss [5,6]. Such molecular deficiencies contribute to tumor immune evasion and represent a major bottleneck limiting the efficacy of immunotherapy.

Emerging evidence has demonstrated a close relationship between autophagy and MHC-I degradation [7,8,9]. As a highly conserved intracellular degradation pathway, autophagy can significantly affect the stability and expression of MHC-I. In particular, hyperactivated autophagy promotes aberrant degradation of MHC-I–peptide complexes, resulting in reduced surface MHC-I levels on tumor cells. This impairs antigen presentation and facilitates immune escape [10]. Therefore, targeting autophagic mechanisms to upregulate MHC-I expression in melanoma cells presents a promising strategy to overcome current therapeutic limitations. While hydroxychloroquine (HCQ) remains the sole FDA-approved autophagy inhibitor in clinical use, its dose-dependent retinopathy (prevalence reaching 7.5%) has raised worldwide therapeutic concerns [11]. Similarly, targeted therapies (such as ATG5/7 knockdown [12] or ULK1/ULK2 inhibitors [13]), while capable of upregulating MHC expression, may lead to unexpected side effects during treatment due to off-target effects, thereby limiting their clinical applications [14]. These limitations underscore the need for agents capable of selectively blocking pathological autophagy without compromising physiological processes.

Encouragingly, natural compound screening has identified Cepharanthine (CEP) as a compound with notable antitumor potential [15]. CEP is a bisbenzylisoquinoline alkaloid extracted from *Stephania japonica* (Thunb.) Miers. Previous studies have identified several antitumor mechanisms of CEP, including: induction of apoptosis via caspase activation [16], reactive oxygen species (ROS) generation [17], modulation of amino acid metabolism [18]; induction of cell cycle arrest, including G1/S phase arrest and DNA damage [19]; and inhibition of angiogenesis via suppression of VEGF signaling [20]. Moreover, CEP has been identified as a P-glycoprotein inhibitor, capable of reversing multidrug resistance and enhancing chemosensitivity [21]. These properties have led to its inclusion in preclinical studies for various malignancies, including leukemia [22] and thrombocytopenia [23].

Given the pivotal role of the autophagy–MHC-I axis in tumor immune evasion and the clinical limitations of current autophagy inhibitors, this study aims to investigate whether CEP can enhance the immunogenicity of melanoma by specifically modulating the autophagy pathway, thereby promoting CD8^+^ T cell-mediated tumor recognition and clearance, while synergistically enhancing the efficacy of PD-1 immune checkpoint blockade. Validation of these hypotheses will provide a theoretical foundation for developing novel combination immunotherapy strategies based on autophagy regulation.

## 2. Materials and Methods

### 2.1. Cell Culture

The murine melanoma cell line B16-F10 (B16) was obtained from the American Type Culture Collection (ATCC, Rockville, MD, USA), while the human melanoma cell line A375 was purchased from Procell Life Science & Technology Co., Ltd. (Wuhan, China). The ovalbumin (OVA)-transfected murine melanoma cell line B16-F10/OVA (B16-OVA) was acquired from Beijing Crispr Biotechnology Co., Ltd. (Beijing, China). All cell lines were maintained at 37 °C in a humidified incubator with 5% CO_2_. B16 and B16-OVA cells were cultured in RPMI-1640 complete medium (RPMI-1640 basal medium supplemented with 10% fetal bovine serum [FBS] and 1% penicillin-streptomycin). A375 cells were maintained in DMEM complete medium (DMEM basal medium containing 10% FBS and 1% penicillin-streptomycin). Their GFP-LC3-expressing counterparts (A375-GFP-LC3) were cultured under identical conditions. Autophagy induction was achieved through nutrient deprivation using Hank’s Balanced Salt Solution (HBSS) for 6 h after thorough phosphate-buffered saline (PBS) washing.

### 2.2. Reagents and Antibodies

Cepharanthine (CEP, purity ≥ 99%, #T0131) was obtained from TargetMol Chemicals (Shanghai, China). The autophagy inhibitors Bafilomycin A1 (#S1413) and Chloroquine (#C6628) were purchased from Selleckchem (Houston, TX, USA). Hank’s Balanced Salt Solution (HBSS, #EH80268) was procured from Taylor Biotechnology (Guangzhou, China). For flow cytometry experiments, the following antibodies were used: PE anti-mouse H-2Kb (#12-5958-82) and PE anti-mouse OVA^257–264^ (SIINFEKL) peptide bound to H-2Kb (#12-5743-82) which were purchased from Invitrogen (Waltham, MA, USA). APC-Cy7 anti-mouse CD45 (#557659), BV510 anti-mouse CD3e (#563024), Fixable Viability Stain 620 (FVS620, #553142), and relevant IgG isotype control antibodies were obtained from BD Biosciences (San Jose, CA, USA). APC anti-mouse CD8a (#100712) was obtained from Biolegend (San Diego, CA, USA). The following antibodies were used for Western blot experiments: β-actin (#3700), LC3-I/II (#12741), p62 (#88588), Cathepsin B (#31718), Cathepsin D (#2284), and MHC-I (#35923) were sourced from Cell Signaling Technology (Danvers, MA, USA). H-2Kb (mouse MHC class I) (#sc-59199) were sourced from Santa Cruz Biotechnology (Dallas, TX, USA). HLA-A/B (#A8754) was provided by ABclonal (Wuhan, China). Secondary antibodies, including HRP-conjugated anti-mouse IgG (#AS003), anti-rabbit IgG (#AS014), and anti-rat IgG (#AS028) were also obtained from ABclonal (Wuhan, China). The anti-CD8a Monoclonal antibody (#14-0081-85) was purchased from Invitrogen (Waltham, MA, USA). Anti-mouse CD8α-InVivo (#A2102) and anti-mouse PD-1 (CD279)-InVivo (#A2122) were provided by Selleckchem (Houston, TX, USA). Anti-rabbit (H+L) (Alexa Fluor^®^ 488, #ab150081) and anti-rat IgG (H+L) (Alexa Fluor^®^ 594, #ab150160) were obtained from Abcam (Cambridge, UK).

### 2.3. Western Blot Analysis

The cellular samples were collected and homogenized in a suitable quantity of ice-cold RIPA lysis buffer containing both protease and phosphatase inhibitor cocktails. Following a 30-min incubation period on ice, the homogenates were subjected to high-speed centrifugation to eliminate insoluble cellular components. The clarified supernatants were subsequently collected, and total protein concentrations were quantified employing a BCA protein assay kit (#MA0082, Meilunbio, Dalian, China). Protein aliquots of equal quantity were resolved through 12% SDS-PAGE and subsequently electrotransferred onto PVDF membranes. After blocking with 5% non-fat dry milk prepared in TBST for 60 min at ambient temperature, the membranes were probed with specific primary antibodies at 4 °C overnight. Following extensive washing, the membranes were incubated with horseradish peroxidase (HRP)-labeled secondary antibodies diluted at 1:5000 for one hour at room temperature. Protein detection was performed using a chemiluminescent substrate system prepared by mixing equal volumes of oxidizing reagent and luminol solution immediately before application. The substrate mixture was uniformly distributed across the membrane surface and allowed to react for 60 s under dark conditions. The resulting chemiluminescent signals were documented using a Tanon 5200CE imaging system (Tanon, Shanghai, China). Quantitative analysis of band intensity was conducted through densitometry with GelPro analysis software 4.0, using β-actin expression as an internal reference for data normalization. To ensure experimental reliability, all procedures were replicated independently in triplicate.

### 2.4. Flow Cytometry Analysis

Murine or human melanoma cells were treated with CEP at the indicated concentrations, harvested by trypsinization with minimal agitation, and the digestion was immediately neutralized using complete medium containing 10% FBS. The cells were subsequently washed with staining buffer (PBS supplemented with 2% FBS) prior to further analysis. Cell pellets were resuspended in antibody working solution (prepared in staining buffer, PBS with 2% FBS) at a 1:100–1:200 dilution, and incubated for 30 min at 4 °C in the dark. After staining, cells were washed, resuspended in staining buffer, and filtered through a 70 μm cell strainer before flow cytometry analysis. Flow cytometric analysis was performed using a BDLSR Fortessa™ flow cytometer (BD Biosciences), with a minimum of 10,000 viable events collected per sample. Data were analyzed using FlowJo software (v10.8.1; Tree Star Inc., Ashland, OR, USA). For single-color fluorescence analysis, IgG isotype controls were used as gating references to define positive cell populations. For tumor cell death quantification, the gating strategy was as follows: forward/side scatter (FSC/SSC) → singlet discrimination → tumor cell identification (CD45^−^). Within the CD45^−^ population, the mean fluorescence intensity (MFI) of FVS620 staining was measured to reflect the extent of tumor cell death. To evaluate the efficiency of in vivo CD8a depletion, splenocytes were isolated from both anti-CD8a-treated and isotype control-treated mice. The gating strategy for CD8^+^ T cell identification included the following steps: singlet selection → CD45^+^ leukocyte gating → CD3^+^CD8^+^ T cell identification. For data analysis, positive gates were defined based on isotype controls. MFI was measured for both vehicle-treated (control) and compound-treated groups. Higher MFI values indicate elevated levels of protein expression. In each experiment, the MFI of the control group was set as the baseline, and relative MFI changes were calculated for the treatment groups. All flow cytometry assays were independently repeated at least three times to ensure data reliability and reproducibility.

### 2.5. In Vitro Cytotoxicity Assay

OT-1 transgenic mice (Strain #GAP2013), harboring a T cell receptor specifically recognizing the OVA^257–264^/H-2K^b^ antigen complex, were procured from GeneEasy Biotech (Yangzhou, China). Splenic cell suspensions were prepared through mechanical dissociation, followed by lymphocyte isolation using density gradient centrifugation (Cat# 7211011, Dakewe Biotech, Shenzhen, China). CD8^+^ T lymphocytes were purified through magnetic separation after incubation with anti-mouse CD8a MicroBeads (Cat# 130-117-044, Miltenyi Biotec) in MACS buffer (Cat# 130-091-222, Miltenyi Biotec, Bergisch Gladbach, Germany) at 4 °C for 10 min. The isolated CD8^+^ T cells were maintained in complete medium containing recombinant IL-2 (Cat# 212-12-100, PeproTech, Cranbury, NJ, USA) for subsequent functional analyses. To evaluate cytotoxic activity, CEP-treated B16-OVA tumor cells were incubated with OT-1-derived CD8^+^ T cells at a target-to-effector ratio of 10:1 for 24 h. Post-co-culture, cellular viability was determined by flow cytometry using CD45 surface staining combined with FVS620 viability dye. Flow cytometric analysis was used to assess tumor cell death by measuring the mean fluorescence intensity (MFI) of FVS620 within the CD45^−^ tumor cell population. This gating strategy enabled the quantification of cell death as an indicator of CD8^+^ T cell cytotoxic activity against CEP-pretreated tumor cells. Relative FVS620 MFI values were compared across treatment conditions to evaluate the extent of CEP-enhanced, T cell-mediated cytotoxicity.

### 2.6. Crystal Violet Cytotoxicity Assay

Following co-culture of CEP-pretreated B16-OVA cells with CD8^+^ T cells, cytotoxic activity was evaluated through crystal violet staining. The experimental procedure involved washing the co-cultured cells twice with PBS to remove non-adherent cells and debris, followed by fixation with 4% paraformaldehyde (PFA) for 15 min at room temperature. Fixed cells were then stained with 0.5% crystal violet solution for 20 min, allowing the dye to bind cellular DNA and proteins as an indicator of viable cell density. After thorough rinsing with distilled water to remove unbound stain, the plates were air-dried prior to quantification. Cell viability was determined by solubilizing the bound dye with 1% acetic acid under gentle agitation for 10 min, followed by absorbance measurement at 595 nm using a microplate reader. The absorbance values, which correlated directly with viable cell numbers, provided a quantitative measure of CD8^+^ T cell-induced cytotoxicity against CEP-pretreated B16-OVA cells.

### 2.7. ELISA for IFN-γ Detection

To measure IFN-γ production, supernatants from co-cultures of CEP-treated B16-OVA cells and CD8^+^ T cells were harvested and clarified by centrifugation at 1500× *g* for 5 min. The supernatants were then diluted 1:50 in assay buffer and transferred to anti-IFN-γ antibody-coated 96-well plates for a 2-h incubation at ambient temperature. After washing, plates were incubated with biotin-conjugated detection antibody for 1 h at room temperature, followed by streptavidin-HRP incubation for 30 min under light-protected conditions. Colorimetric development was initiated by substrate addition for 20 min in darkness before reaction termination. Absorbance measurements at 450 nm were performed immediately using a plate reader, with IFN-γ concentrations determined by normalizing sample optical density values to those obtained from OT-1 T cell monoculture controls.

### 2.8. GFP-LC3 Autophagosome Detection

To monitor autophagosome formation, a stable A375 melanoma cell line expressing a GFP-LC3 fusion protein was established. In this system, autophagosomes are visualized as distinct green fluorescent puncta. A375-GFP-LC3 cells were seeded at an appropriate density onto sterile glass coverslips and allowed to adhere overnight. Cells were then treated with various concentrations of test compounds for 24 h. Following treatment, cells were fixed with 4% paraformaldehyde for 15 min at room temperature and counterstained with 4′,6-Diamidino-2-Phenylindole (DAPI, 5 μg/mL in PBS) for 10 min to visualize nuclei. After thorough washing with PBS, samples were mounted and imaged using a confocal microscope equipped with a 63× oil immersion objective. For each group, images from 10 randomly selected fields were captured. Autophagic activity was quantified by counting the number of GFP-LC3 puncta per cell. The average number of puncta per cell was calculated as a quantitative indicator of autophagosome accumulation and autophagy induction.

### 2.9. mCherry-GFP-LC3 Autophagic Flux Assay

Autophagic flux was evaluated using the mCherry-GFP-LC3 dual-fluorescence reporter system, which takes advantage of the pH sensitivity of GFP fluorescence—quenched in acidic lysosomes—while mCherry remains stable. This enables simultaneous visualization of autophagosomes and autolysosomes, thus providing insight into the dynamic process of autophagy. Tumor cells were transfected with the pBABE-puro-mCherry-GFP-LC3B plasmid (Addgene, Cat# 22418), followed by CEP treatment after 24 h. After treatment, cells were fixed with 4% paraformaldehyde, counterstained with DAPI and imaged using an LSM 800 confocal microscope (Carl Zeiss, Jena, Germany) equipped with a 63× oil immersion objective. Excitation wavelengths were set at 488 nm (GFP), 561 nm (mCherry), and 405 nm (DAPI). For each experimental condition, 10–15 randomly selected fields were analyzed. In this system, yellow puncta (GFP^+^/mCherry^+^) represent autophagosomes prior to fusion with lysosomes, while red puncta (mCherry^+^ only) indicate autolysosomes, in which GFP fluorescence is quenched due to the acidic lysosomal environment. Accumulation of yellow puncta (GFP^+^/mCherry^+^) may reflect impaired autophagic flux due to blocked fusion or defective lysosomal acidification. This assay provides a reliable approach to assessing the impact of experimental treatments on autophagic progression.

### 2.10. Immunofluorescence Detection of Autophagosome–Lysosome Fusion

Immunofluorescence staining was performed to assess the colocalization of lysosomal membrane protein LAMP1 (red fluorescence, mCherry-tagged) and the autophagosome marker LC3 (green fluorescence) in A375-hLAMP1-mCherry/LC3 cells, thereby enabling direct visualization of autophagosome–lysosome fusion events. A375-hLAMP1-mCherry cells were seeded on glass coverslips and exposed to specified drug concentrations for 24 h. After treatment, cellular samples were fixed using 4% paraformaldehyde, permeabilized with 0.1% Triton X-100 in PBS for 10 min at room temperature, and subsequently blocked with 5% BSA for 1 h. For immunofluorescence detection, fixed cells were first probed with LC3B-specific primary antibodies (rabbit polyclonal) at 4 °C overnight, then labeled with Alexa Fluor 488-tagged secondary antibodies under light-protected conditions for 60 min at room temperature to visualize autophagosome formation. After extensive washing, nuclei were counterstained with DAPI (5 min, light-protected), and coverslips were mounted using antifade mounting medium. Imaging was performed using a Zeiss LSM 800 confocal microscope equipped with a 63× oil immersion objective. The following excitation wavelengths were used: 405 nm (DAPI), 488 nm (Alexa Fluor 488), and 561 nm (mCherry). For each condition, 8–10 randomly selected fields were imaged.

### 2.11. Evaluation of Lysosomal Acidification Using LysoTracker

Lysosomal PH was evaluated using LysoTracker Red, a fluorescent probe that specifically targets acidic compartments. A375 cells were cultured in confocal dishes and treated with the indicated test compounds for 24 h. After treatment, cells were incubated with LysoTracker Red (50 nM; Thermo Fisher Scientific, Waltham, MA, USA, Cat# L7528) for 20 min at 37 °C under light-protected conditions. Following incubation, cells were washed gently with PBS to remove excess dye. Fluorescence signals indicating lysosomal acidification were captured using confocal microscopy, and signal intensity was quantified to evaluate changes in lysosomal pH under different treatment conditions.

### 2.12. Immunofluorescence Staining

Paraffin-embedded tissue sections were sequentially processed through xylene dewaxing, ethanol rehydration, and heat-mediated antigen retrieval in citrate buffer (pH 6.0). Non-specific binding sites were blocked with 5% bovine serum albumin (BSA) for 1 h at room temperature, followed by incubation with anti-CD8 primary antibody (diluted 1:200 in PBS) at 4 °C overnight. After thorough PBS washes, sections were exposed to Alexa Fluor^®^ 594-conjugated anti-rat IgG secondary antibody (Abcam, catalog #ab150160) for 60 min under light-protected conditions and then counterstained with DAPI. Fluorescent signals were captured with a Zeiss LSM 800 laser scanning confocal microscope (Carl Zeiss AG, Jena, Germany) and analyzed using ImageJ (1.53t).

### 2.13. Animal Tumor Model

All procedures involving animals were performed in compliance with the ethical standards approved by the Institutional Animal Care and Use Committee (IACUC) at Guangzhou University of Chinese Medicine (Approval No. 20240528002). Male C57BL/6 mice (8 weeks old, SPF-grade), obtained from Guangdong Provincial Medical Laboratory Animal Center, were maintained in a regulated environment with ad libitum access to autoclaved feed and water. For tumor induction, each mouse received a subcutaneous injection of 2 × 10^6^ B16 melanoma cells (in 100 μL PBS) into the right flank region. Treatment protocols were initiated 24 h after tumor inoculation. Mice were then randomly assigned to one of three treatment groups (n = 5 per group): vehicle control (saline), CEP at 20 mg/kg, and CEP at 40 mg/kg. All treatments were administered once daily via intraperitoneal injection (i.p.). To investigate the mechanistic role of CD8^+^ T cells, an additional set of mice were assigned to four experimental groups: Control, Anti-CD8 antibody (αCD8; 200 μg/mouse, i.p. every 72 h), CEP alone (40 mg/kg, i.p. daily), and CEP + αCD8 combination therapy. In parallel, the synergistic effect of PD-1 blockade was evaluated using the following treatment groups: Control, Anti–PD-1 antibody (αPD-1; 200 μg/mouse, i.p. every 72 h), CEP alone (40 mg/kg, i.p. daily), and CEP + αPD-1 combination therapy. Mice were monitored until tumor volumes approached the humane endpoint (~800 mm^3^), at which point they were euthanized, and tumors were excised, weighed, and analyzed.

### 2.14. Statistical Analysis

Statistical evaluations were carried out with GraphPad Prism version 6.0 (GraphPad Software, San Diego, CA). Numerical results are expressed as mean ± SD or SEM, as specified in figure legends. Two-group comparisons employed Student’s *t*-test, while multi-group analyses utilized either one-way or two-way ANOVA, with subsequent post hoc testing (Dunnett’s or Tukey’s test) where applicable. Statistical significance was defined as follows: *p* < 0.05 (*), *p* < 0.01 (**), and *p* < 0.001 (***). Results not reaching statistical significance were reported as “ns” (not significant). All experiments were repeated independently at least three times to ensure reproducibility.

## 3. Results

### 3.1. CEP Enhances MHC-I-Mediated Antigen Presentation in Melanoma Cells

Cepharanthine (CEP), a bisbenzylisoquinoline alkaloid derived from *Stephania japonica* (Thunb.) Miers of the Menispermaceae family (Figure 1A), was investigated for its immunomodulatory effects on major histocompatibility complex class I (MHC-I) expression. Given the well-established role of MHC-I downregulation in tumor immune evasion [5,6], we assessed CEP’s ability to modulate MHC-I expression in melanoma models. Two cell lines were employed: human A375 cells expressing HLA-A/B and murine B16 cells expressing H-2K^b^. Following 24 h of treatment with increasing concentrations of CEP (0, 1.25, 2.5, and 5 μM), flow cytometric analysis revealed a dose-dependent upregulation of surface MHC-I expression in B16 cells (Figure 1B). Consistently, Western blot analysis demonstrated elevated intracellular MHC-I protein levels in both A375 and B16 cells (Figure 1C). To determine the functional consequences of MHC-I upregulation, we employed two antigen presentation models: wild-type B16 cells pulsed with the OVA^257–264^ peptide and B16-OVA cells stably expressing ovalbumin. Flow cytometry showed that CEP treatment led to a dose-dependent increase in surface expression of the H-2K^b^/SIINFEKL complex in both models (Figure 1D,E), indicating enhanced antigen presentation capacity. Together, these findings demonstrate that CEP enhances the antigen presentation machinery of melanoma cells by upregulating MHC-I expression at both the cell surface and total protein levels. This dual effect highlights CEP as a novel immuno-modulatory compound capable of boosting melanoma cell immunogenicity, thereby holding promise for enhancing the effectiveness of immune-based therapeutic approaches against melanoma.

### 3.2. CEP Enhances CD8^+^ T Cell-Mediated Killing of Melanoma Cells

Having established CEP’s role in augmenting MHC-I antigen processing in malignant melanocytes, subsequent experiments were designed to characterize its immunomodulatory effects on cytotoxic T cell-mediated tumor cell lysis. To exclude the possibility of direct cytotoxic effects, we first performed CCK-8 viability assays, which confirmed that CEP did not significantly affect melanoma cell viability at the tested concentrations (Appendix A). We then conducted co-culture assays in which B16-OVA melanoma cells were pretreated with CEP for 24 h, followed by co-incubation with OT-1 CD8^+^ T cells (Figure 2A). Notably, CEP was applied exclusively to tumor cells, enabling us to specifically assess its impact on tumor immunogenicity and antigen presentation, rather than direct modulation of T cell function. Flow cytometric analysis revealed a significant, dose-dependent increase in tumor cell apoptosis following CEP pretreatment (Figure 2B,C), indicating enhanced susceptibility of tumor cells to CD8^+^ T cell-mediated killing. This effect is attributable to CEP-induced upregulation of antigen presentation, which likely facilitates improved T cell recognition and activation. Consistent with this, crystal violet staining demonstrated that CEP exhibited no significant effect on tumor cell viability in the absence of CD8^+^ T cells. However, upon CD8^+^ T cell co-culture, the number of surviving tumor cells showed an inverse correlation with CEP concentration (Figure 2D). IFN-γ secretion was significantly elevated in co-culture supernatants from CEP-treated groups, as measured by ELISA (Figure 2E, further supporting the notion that CEP amplifies T cell effector function by enhancing tumor cell immunogenicity. The cumulative data reveal that CEP’s immunomodulatory action operates through MHC-I pathway potentiation, which enhances tumor antigen visibility to cytotoxic T lymphocytes without inducing direct tumor cell death. This highlights CEP as a promising non-cytotoxic immunomodulatory agent capable of improving the efficacy of T cell-based immunotherapies by strengthening the antigen presentation–cytotoxicity axis within the tumor microenvironment.

### 3.3. CEP Upregulates MHC-I Levels and Antigen Presentation Through Autophagy Blockade in Melanoma Cells

Our initial findings demonstrated that CEP treatment increases MHC-I expression in melanoma cells. Given the well-established role of autophagy in mediating MHC-I degradation, we further explored whether CEP exerts its immunomodulatory effects via autophagy inhibition. To this end, we employed Bafilomycin A1 (Baf), a classical autophagy inhibitor, as a pharmacological reference. B16 cells were exposed to increasing concentrations of Baf (0, 25, 50, and 75 nM) for 24 h. Flow cytometric analysis revealed a dose-dependent upregulation of surface MHC-I expression (Figure 3A). This elevation in antigen presentation machinery was further validated at the protein level through Western blot analysis, with both B16 and A375 cells exhibiting progressive increases in total MHC-I expression that correlated with Baf concentration (Figure 3B). In addition, Baf treatment led to a dose-dependent increase in the surface presentation of H-2K^b^/SIINFEKL complexes, as detected by flow cytometry (Figure 3C), indicating enhanced antigen presentation. To determine whether CEP and Baf act through similar pathways, we performed combination treatment experiments. The results showed non-additive effects on MHC-I expression (Figure 3D,E), suggesting that CEP and Baf share a common mechanism—namely, autophagy blockade—to enhance MHC-I-mediated antigen presentation in melanoma cells.

### 3.4. CEP Functions as an Autophagy Inhibitor in Melanoma Cells

To investigate the impact of CEP on autophagic dynamics, we utilized A375-GFP-LC3 melanoma cells to visualize autophagosome formation in real time via fluorescence-based tracking. To delineate the underlying mechanism, we included parallel control groups treated with HBSS (a classical autophagy inducer), as well as chloroquine (CQ) and bafilomycin A1 (Baf), both well-established autophagy inhibitors. Confocal microscopy analysis revealed a marked increase in GFP-LC3 puncta formation in cells treated with HBSS, Baf, CQ, or CEP, compared to vehicle-treated controls (Figure 4A), indicating substantial autophagosome accumulation. Notably, CEP induced autophagosome accumulation to a similar extent as CQ and Baf, suggesting a comparable inhibitory effect on autophagic flux. To further validate this observation, we examined the expression of canonical autophagy markers LC3-II and p62, whose co-accumulation is indicative of blocked autophagic degradation [24]. Western blot analysis demonstrated progressive accumulation of LC3-II and p62 in both A375 and B16 cells upon exposure to escalating doses of CEP (Figure 4B). Time-course analysis revealed that this effect was progressively amplified over 24 h (Figure 4C), further supporting sustained inhibition of autophagic flux. Strikingly, co-treatment with CEP and HBSS resulted in a synergistic enhancement of autophagosome accumulation (Figure 4D), reinforcing the conclusion that CEP acts as an inhibitor of autophagic degradation, rather than merely an inducer of autophagosome formation. Together, these findings demonstrate that CEP inhibits autophagy in melanoma cells, leading to the accumulation of autophagic vesicles, which likely contributes to its immunomodulatory effects via stabilization of MHC-I molecules.

### 3.5. CEP Blocks Autophagic Flux via Lysosomal Acidification Inhibition

Autophagy is a highly conserved intracellular degradation process that maintains cellular homeostasis under stress conditions [25]. During autophagy initiation, stimuli such as nutrient deprivation or oxidative stress promote autophagosome formation, which subsequently fuses with lysosomes to form autolysosomes. Within autolysosomes, lysosomal acid hydrolases (e.g., cathepsins) degrade the engulfed contents, releasing small molecules such as amino acids and free fatty acids for cellular recycling. To elucidate the precise mechanism by which CEP impairs autophagic flux, we first established an A375-mCherry-GFP-LC3 dual-fluorescent reporter cell line, which enables real-time discrimination between autophagosomes (yellow fluorescence) and autolysosomes (red fluorescence only). Confocal microscopy revealed that, similar to chloroquine (CQ) treatment, CEP exposure resulted in persistent yellow puncta due to the retention of GFP fluorescence (Figure 5A), suggesting impaired autophagosome–lysosome fusion or lysosomal acidification.

To distinguish between these possibilities, we next generated A375-mCherry-hLAMP1-LC3 cells to directly visualize autophagosome–lysosome fusion via colocalization of the lysosomal membrane marker LAMP1 (red) and autophagosome marker LC3 (green). Notably, CEP treatment did not disrupt LAMP1–LC3 colocalization, indicating that autophagosome–lysosome fusion remains intact (Figure 5B). To evaluate lysosomal function, we performed LysoTracker staining and observed that CEP treatment markedly diminished lysosomal acidification, mirroring the effect of bafilomycin A1 (Baf) (Figure 5C). Subsequent investigation of lysosomal protease processing revealed that CEP treatment impaired the proteolytic maturation of both cathepsin B and cathepsin D in a concentration- and duration-dependent manner across melanoma cell lines (Figure 5D,E). These results establish that CEP primarily compromises autophagic degradation through lysosomal pH dysregulation, which subsequently disrupts the acidic microenvironment essential for lysosomal enzyme activation, while leaving the autophagosome–lysosome fusion process unaffected.

### 3.6. CEP Suppresses Melanoma Growth via CD8^+^ T Cell-Mediated Immunity and Synergizes with Anti-PD-1 Therapy

Building upon our in vitro findings, we next investigated the in vivo antitumor efficacy of CEP using a subcutaneous B16 melanoma mouse model. This allowed us to systematically evaluate both the dose-dependent effects of CEP on tumor progression and its influence on tumor-infiltrating immune cells. In vivo tumorigenicity assays demonstrated that CEP significantly suppressed melanoma growth in a dose-dependent manner. Compared to vehicle-treated controls, both 20 mg/kg and 40 mg/kg CEP treatments resulted in marked tumor growth inhibition, with the higher dose exhibiting superior efficacy (Figure 6A–C). Immunofluorescence analysis of tumor sections revealed a substantial increase in CD8^+^ T cell infiltration in CEP-treated tumors (Figure 6D), suggesting enhanced antitumor immunity.

To determine whether CD8^+^ T cells are essential for the antitumor activity of CEP, we performed CD8^+^ T cell depletion using anti-CD8 antibodies. Flow cytometric analysis of splenocytes confirmed effective and specific depletion of CD8^+^ T cells in vivo, without significantly altering other immune cell populations (Appendix A). Notably, the antitumor activity of CEP was substantially attenuated upon CD8^+^ T cell ablation (Figure 6F–H), highlighting the essential contribution of CD8^+^ T cell-dependent immune responses to its therapeutic effects. Moreover, CEP displayed remarkable combinatorial potential with PD-1 blockade, with the combined regimen producing significantly greater tumor suppression than either treatment in isolation (Figure 6I–K). This synergistic interaction suggests that CEP may serve as an effective immunomodulatory agent to augment the clinical benefits of immune checkpoint inhibition. Together, these data establish that CEP mediates its antimelanoma effects principally via engagement of CD8^+^ T cell immunity, while simultaneously functioning as a promising combination partner for PD-1-targeted immunotherapy to achieve enhanced therapeutic responses.

## 4. Discussion

We demonstrate that CEP effectively inhibits late-stage autophagy, thereby augmenting MHC-I antigen presentation and stimulating CD8^+^ T cell-mediated antitumor responses in melanoma. Mechanistically, CEP disrupts lysosomal acidification, thereby impairing autophagic degradation and leading to the accumulation of MHC-I molecules. This mechanism is distinct from previously reported transcriptional or epigenetic upregulation strategies, such as IFN-γ stimulation or HDAC inhibition. Importantly, CEP significantly increased tumor antigen presentation, as evidenced by elevated H-2K^b^/SIINFEKL complex formation, and enhanced CD8^+^ T cell cytotoxicity both in vitro and in vivo. Furthermore, CEP enhances the efficacy of anti–PD-1 immune checkpoint therapy. These findings position CEP as a novel immunomodulatory agent capable of reversing tumor immune evasion by targeting the autophagy–MHC-I axis.

Compared to conventional autophagy inhibitors, CEP offers unique advantages in specificity and therapeutic potential. Agents such as chloroquine and bafilomycin A1 broadly block autophagy at various stages [26,27], often resulting in widespread cellular effects. In contrast, CEP selectively inhibits lysosomal acidification without affecting autophagosome–lysosome fusion, as demonstrated by our imaging and colocalization analyses (Figure 5B,C). This selectivity may underlie its favorable toxicity profile observed in preclinical models (Figure 6). Moreover, unlike mTOR inhibitors (e.g., rapamycin), which act upstream to suppress autophagy indirectly [28], CEP directly impairs lysosomal function, blocking cathepsin maturation (Figure 5D,E) and effectively preventing MHC-I degradation (Figure 1B,C)—a mechanism increasingly recognized as critical for CD8^+^ T cell activation in melanoma immunotherapy [7]. While prior studies have explored autophagy inhibition to enhance antitumor immunity, most have focused on early-stage autophagy blockade, such as ATG5 knockdown, which may trigger compensatory metabolic pathways. In contrast, our findings highlight the therapeutic superiority of late-stage autophagy inhibition. CEP not only augments antigen presentation (Figure 1D,E) but also demonstrates robust synergy with PD-1 checkpoint blockade (Figure 6I–K), achieving superior tumor control compared to monotherapies.

Notably, this study further corroborates the favorable short-term safety profile of CEP. In vitro experiments revealed no significant direct cytotoxicity (Appendix A) or adverse effects on tumor cell viability (Figure 2D) at therapeutic concentrations, supporting its low toxicity upon short-term exposure. In our in vivo studies, intraperitoneal administration of CEP (20–40 mg/kg) over several weeks did not induce substantial body weight loss or behavioral abnormalities, indicating minimal acute toxicity. However, it is well-established that autophagy inhibitors may disrupt tissue homeostasis [29], particularly in organs dependent on autophagy for protein and organelle clearance, such as the liver [30] and nervous system [31]. While this study did not explicitly assess long-term toxicity, structurally related alkaloids (e.g., berbamine derivatives [32]) exhibit low cumulative toxicity, suggesting a potentially favorable profile for CEP. Nevertheless, further investigations—including repeated-dose toxicity, reproductive toxicity, and pharmacokinetic studies—are essential to define its therapeutic window and facilitate clinical translation.

The clinical implications of these findings are significant, particularly for addressing MHC-I downregulation—a major mechanism of immune escape in advanced melanoma. Over 60% of metastatic melanomas exhibit reduced MHC-I expression, limiting the efficacy of checkpoint inhibitors [33]. CEP’s ability to restore MHC-I levels without genetic modification offers a pharmacologically tractable approach to overcoming this barrier. Additionally, the observed increase in intratumoral CD8^+^ T cells (Figure 6D) suggests that CEP may convert immune “cold” tumors into “hot” ones, thereby expanding the population of patients who could benefit from immunotherapy. Although this study primarily focused on melanoma, the conserved role of autophagy in MHC-I regulation across multiple cancers (e.g., non-small-cell lung cancer [34] and colorectal cancer [35]) suggests that CEP may also hold therapeutic potential for other MHC-I-deficient, autophagy-dependent malignancies. Future studies should explore its efficacy in these contexts and evaluate synergistic opportunities with emerging immunotherapies, such as TIGIT/LAG-3 blockade [36].

Despite the promising results, several limitations warrant further investigation. First, although we have clearly demonstrated that CEP inhibits lysosomal acidification, its precise molecular target remains unidentified. The existing literature suggests that the V-type ATPase plays a pivotal role in tumor lysosomal acidification [37], potentially serving as a target of CEP. This hypothesis requires validation through future proteomic analysis and molecular docking experiments. Second, while CEP exhibited favorable pharmacokinetic properties in murine models, quantitative data on systemic exposure and tumor tissue penetration are currently lacking. This limitation primarily stems from the unavailability of radiolabeled CEP, prompting our ongoing development of an LC-MS/MS-based detection method to address this gap. Furthermore, although the B16 and A375 models employed in this study are well-established, their genomic profiles do not fully recapitulate the heterogeneity of human melanoma [38]. Notably, the absence of models harboring BRAF, CDKN2A, NRAS, or TP53 mutations may limit the generalizability of our findings. We plan to incorporate patient-derived xenograft (PDX) models in subsequent studies to overcome this limitation.

In conclusion, our study establishes CEP as a late-stage autophagy inhibitor that enhances antitumor immunity by disrupting lysosomal pH homeostasis and stabilizing MHC-I expression. Its ability to synergize with PD-1 blockade and its favorable safety profile support its clinical translation as a novel immunotherapeutic agent. These findings not only deepen our understanding of autophagy–immune crosstalk, but also provide a promising strategy to overcome immune evasion in melanoma and potentially other cancers. Future research should focus on refining CEP’s molecular mechanism, optimizing its delivery, and expanding its application to next-generation combination immunotherapies.

## 5. Conclusions

Our study demonstrates that CEP enhances MHC-I-mediated antigen presentation in melanoma by inhibiting lysosomal acidification, thereby blocking autophagy-dependent MHC-I degradation. This mechanism promotes CD8^+^ T cell-mediated tumor clearance and synergizes with anti–PD-1 therapy to improve antitumor immunity. Unlike conventional autophagy inhibitors, CEP selectively targets late-stage autophagy without interfering with autophagosome–lysosome fusion, offering improved specificity and a favorable safety profile. Collectively, these findings establish CEP as a promising immunomodulatory agent capable of overcoming tumor immune evasion by targeting the autophagy–MHC-I axis. Further studies are warranted to explore its clinical potential, particularly in the context of combination immunotherapies.

## Figures and Tables

**Figure 1 cells-14-01231-f001:**
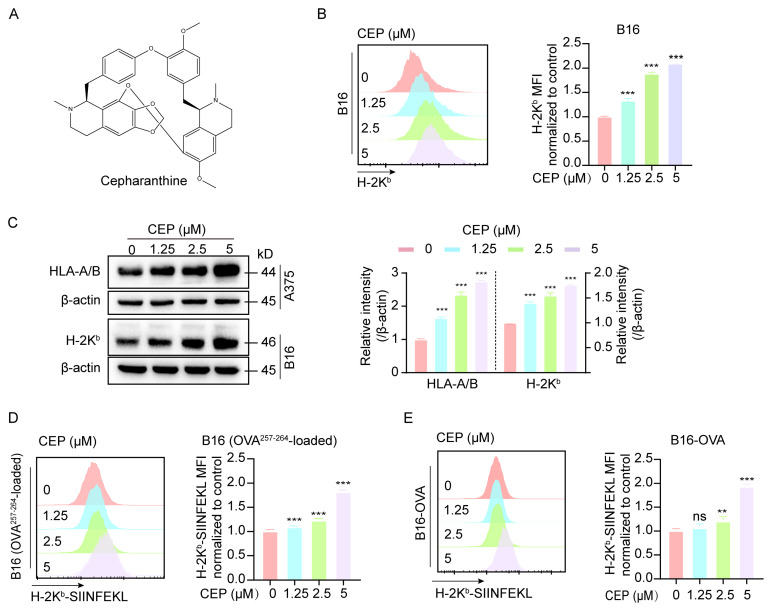
CEP enhances MHC-I-mediated antigen presentation in melanoma cells. (**A**) Chemical structure of cepharanthine (CEP). (**B**) CEP increases surface MHC-I expression in a concentration-dependent manner. B16 melanoma cells were treated with graded concentrations of CEP (0, 1.25, 2.5, 5 μM) for 24 h, followed by flow cytometric analysis using an anti–H-2K^b^ antibody. Representative histograms and normalized mean fluorescence intensity (MFI) are shown. (**C**) CEP elevates total MHC-I protein levels in melanoma cells. B16 and A375 cells were treated with CEP (0, 1.25, 2.5, 5 μM) for 24 h and analyzed by Western blot to detect total MHC-I protein levels (mouse H-2K^b^; human HLA-A/B). (**D**,**E**) CEP enhances MHC-I–restricted antigen presentation. B16 cells pulsed with OVA^257–264^ peptide and B16-OVA cells (stably expressing ovalbumin) were treated with CEP (0, 1.25, 2.5, 5 μM) for 24 h. Surface expression of the H-2K^b^/SIINFEKL complex was quantified by flow cytometry. Data represent mean ± SEM from at least three independent experiments. ** *p* < 0.01 and *** *p* < 0.001 indicate levels of statistical significance.

**Figure 2 cells-14-01231-f002:**
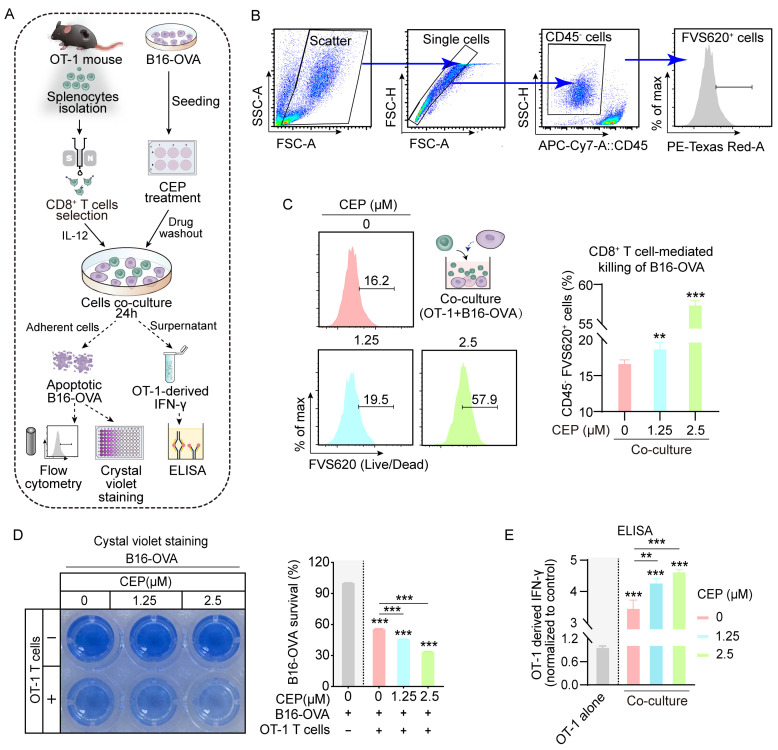
CEP enhances CD8^+^ T cell-mediated cytotoxicity against melanoma cells. (**A**) Schematic of the in vitro immune cytotoxicity assay. B16-OVA cells were pretreated with increasing concentrations of CEP (0, 1.25, 2.5 μM) for 24 h, followed by co-culture with CD8^+^ T cells (isolated from OT-1 mouse spleens) at a 1:10 (target-to-effector) ratio for 24 h. T cell activation and cytotoxicity were evaluated by: (1) flow cytometric detection of apoptotic B16-OVA cells; (2) crystal violet staining of live B16-OVA cells; and (3) ELISA quantification of IFN-γ secretion. (**B**) Gating strategy for quantifying tumor cell death. Flow cytometry gating steps: (1) forward/side scatter; (2) single cells; (3) tumor cell identification (CD45^−^); (4) dead tumor cell identification (CD45^−^FVS620^+^). (**C**) CEP enhances CD8^+^ T cell-mediated killing of tumor cells. Bar graph shows the percentage of apoptotic tumor cells (APC–CD45^−^FVS620^+^) following treatment. (**D**) Crystal violet staining demonstrates that CEP enhances CD8^+^ T cell-mediated killing of B16-OVA cells. Although CEP alone exhibits no intrinsic cytotoxicity, it synergizes with CD8^+^ T cells to eliminate tumor cells in a dose-dependent manner. (**E**) CEP promotes CD8^+^ T cell activation. IFN-γ levels in co-culture supernatants were quantified by ELISA. Data represent mean ± SEM from at least three independent experiments. ** *p* < 0.01 and *** *p* < 0.001 indicate statistical significance. CEP, cepharanthine.

**Figure 3 cells-14-01231-f003:**
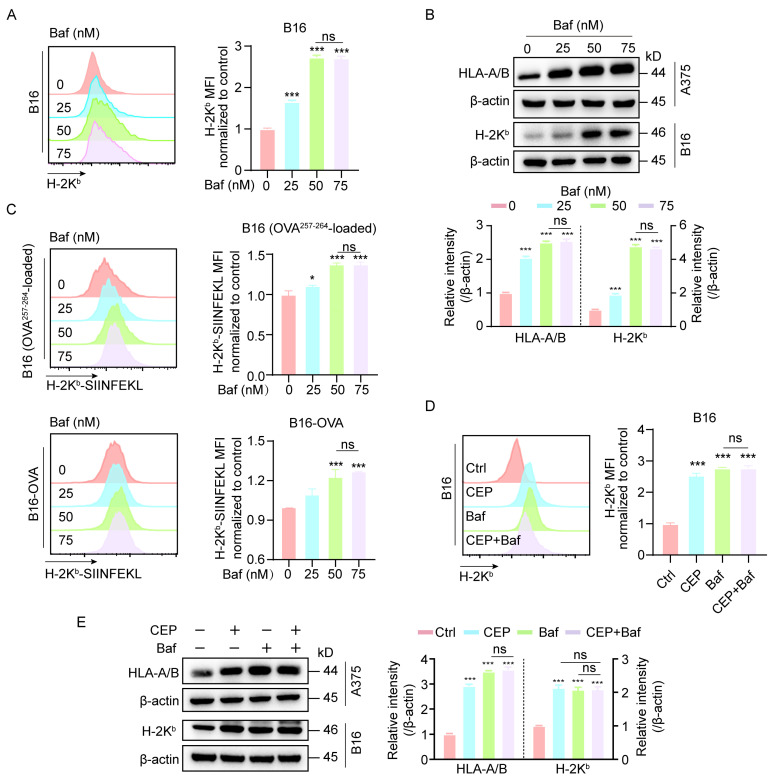
CEP upregulates MHC-I expression in melanoma cells by blocking autophagy. (**A**,**B**) Bafilomycin A1 (Baf), a well-characterized late-stage autophagy inhibitor, increases both surface and total MHC-I expression in melanoma cells. B16 and A375 cells were treated with increasing concentrations of Baf (0, 25, 50, 75 nM) for 24 h. Surface MHC-I (H-2K^b^) expression was assessed by flow cytometry using fluorescent antibody staining, and total MHC-I protein levels were measured by Western blot analysis. (**C**) Autophagy inhibition enhances MHC-I–restricted antigen presentation. B16 cells pulsed with OVA^257–264^ peptide and B16-OVA cells were treated with Baf (0, 25, 50, 75 nM) for 24 h. Surface expression of the H-2K^b^/SIINFEKL complex was quantified by flow cytometry (mean fluorescence intensity). (**D**) Combined treatment with CEP and Baf does not further increase surface MHC-I expression. B16 cells were treated with vehicle control, CEP (5 μM), Baf (50 nM), or a combination of both for 24 h, followed by flow cytometric analysis of surface MHC-I levels. (**E**) Co-treatment with CEP and Baf does not produce additive effects on total MHC-I protein expression. B16 and A375 cells were treated with vehicle, CEP (5 μM), Baf (50 nM), or their combination for 24 h, and total MHC-I levels were assessed via Western blotting. Data represent mean ± SD from at least three independent experiments. * *p* < 0.05 and *** *p* < 0.001 indicates statistical significance. CEP, cepharanthine; Baf, bafilomycin A1.

**Figure 4 cells-14-01231-f004:**
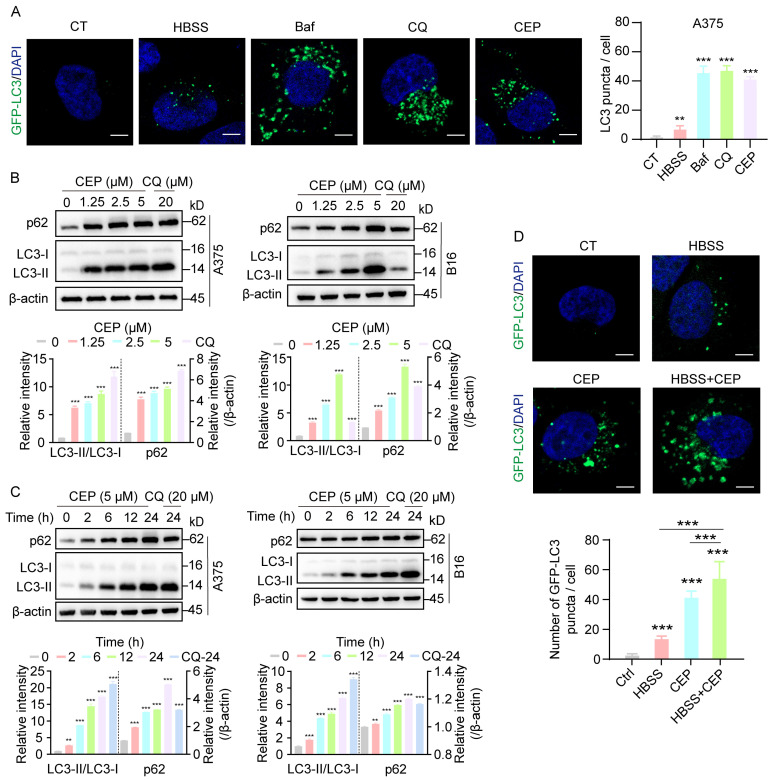
CEP inhibits autophagic flux in melanoma cells. (**A**) CEP promotes autophagosome accumulation in melanoma cells. A375 cells transfected with GFP-LC3 were treated with HBSS (autophagy inducer, 6 h), Bafilomycin A1 (Baf, 20 nM, 24 h), chloroquine (CQ, 20 μM, 24 h), or CEP (5 μM, 24 h). GFP-LC3 puncta were visualized by confocal microscopy and quantified as a measure of autophagosome accumulation. Scale bar: 5 μm. (**B**) CEP induces a dose-dependent accumulation of autophagy markers. A375 and B16 melanoma cells were treated with increasing concentrations of CEP (0, 1.25, 2.5, 5 μM) for 24 h, with CQ (20 μM) as a positive control. Protein levels of p62 and LC3-II were assessed by Western blot. (**C**) CEP increases p62 and LC3-II levels in a time-dependent manner. A375 and B16 cells were treated with CEP (5 μM) for the indicated time points (0, 2, 6, 12, 24 h), with CQ (20 μM) as a positive control, followed by Western blot analysis. (**D**) Combined treatment with CEP and HBSS synergistically enhances autophagosome accumulation. A375-GFP-LC3 cells were treated with vehicle, HBSS (6 h), CEP (5 μM, 24 h), or the combination of HBSS and CEP. GFP-LC3 puncta were analyzed by confocal microscopy. Scale bar: 5 μm. Data represent mean ± SEM from at least three independent experiments. ** *p* < 0.01 and *** *p* < 0.001 indicate statistical significance. CEP, cepharanthine; Baf, bafilomycin A1; CQ, chloroquine.

**Figure 5 cells-14-01231-f005:**
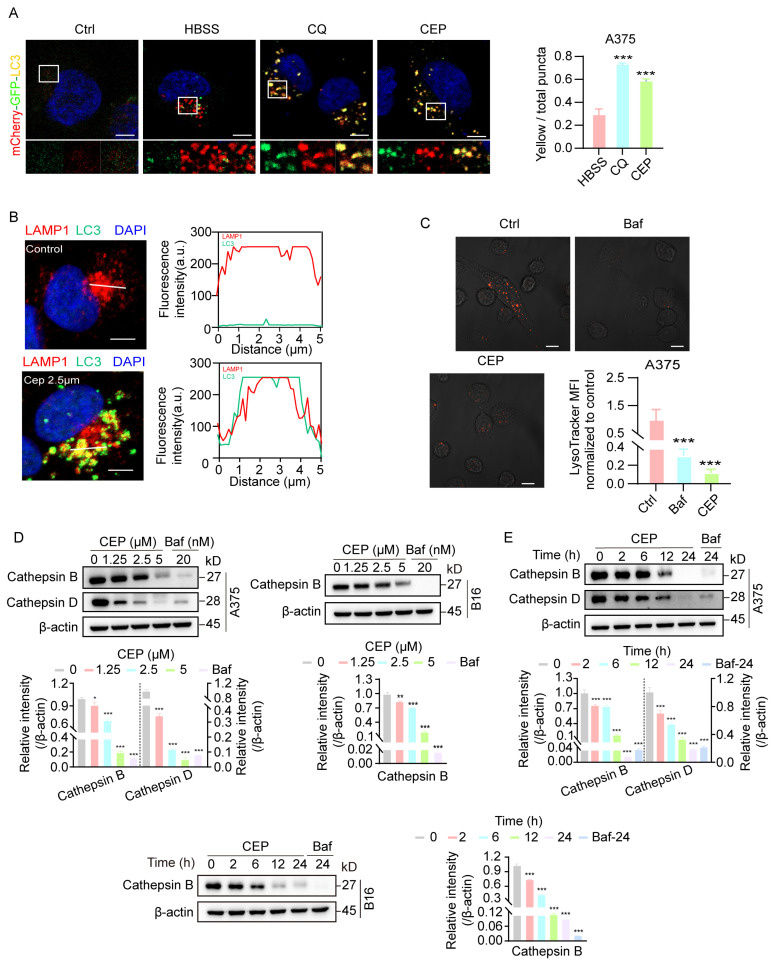
CEP inhibits late-stage autophagic flux by impairing lysosomal acidification without blocking autophagosome–lysosome fusion. (**A**) CEP blocks late-stage autophagic flux in melanoma cells. A375 cells transfected with mCherry-GFP-LC3 were treated with vehicle, HBSS (starvation, 6 h), chloroquine (CQ, 20 μM, 24 h), or CEP (5 μM, 24 h). Confocal microscopy was used to visualize yellow puncta (merged red and green fluorescence), indicative of impaired autophagic flux due to blocked lysosomal degradation. Scale bar: 5 μm. (**B**) CEP does not inhibit autophagosome–lysosome fusion. A375 cells transfected with hLAMP1-mCherry were treated with vehicle or CEP (2.5 μM, 24 h), followed by LC3 immunofluorescence (green). Colocalization between hLAMP1 (red) and LC3 (green) was quantified using ImageJ. Yellow puncta and overlapping intensity profiles (red/green) in magnified regions indicate preserved autophagosome–lysosome fusion. Scale bar: 5 μm. (**C**) CEP impairs lysosomal acidification. A375 cells were treated with vehicle, Baf (20 nM, 24 h), or CEP (5 μM, 24 h), then stained with LysoTracker. Reduced red fluorescence in CEP- and Baf-treated cells indicates loss of lysosomal acidity. Scale bar: 5 μm. (**D**,**E**) CEP inhibits cathepsin maturation in a dose- and time-dependent manner. A375 and B16 cells were treated with either: (1) increasing concentrations of CEP (0, 1.25, 2.5, 5 μM, 24 h), or (2) a fixed concentration of CEP (5 μM) for varying durations (0, 2, 6, 12, 24 h). Western blot analysis was performed to assess cathepsin maturation. Baf (20 nM) was used as a positive control. Data represent mean ± SEM from at least three independent experiments. * *p* < 0.05, ** *p* < 0.01 and *** *p* < 0.001 indicate statistical significance. CEP, cepharanthine; Baf, bafilomycin A1; CQ, chloroquine.

**Figure 6 cells-14-01231-f006:**
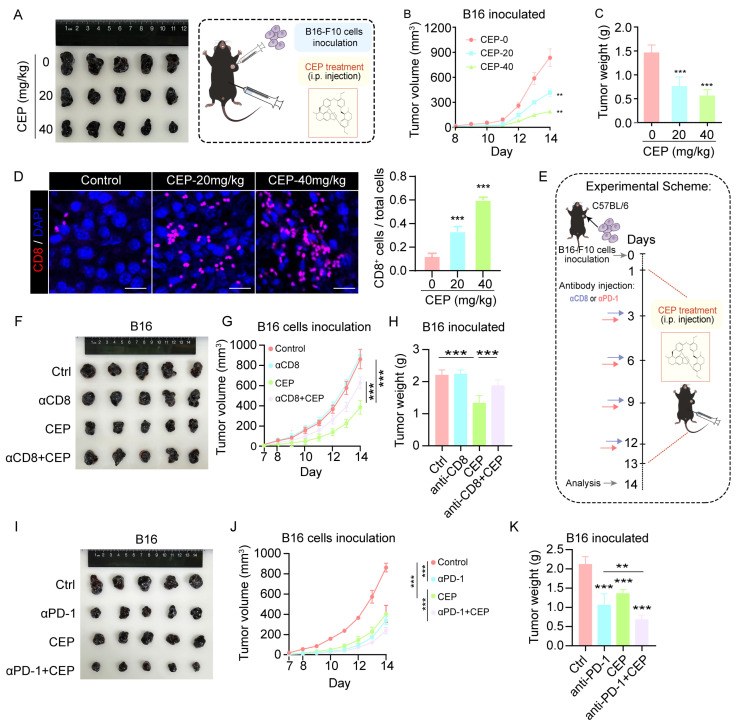
CEP inhibits melanoma growth via CD8^+^ T cell-mediated immunity and enhances anti-PD-1 therapy efficacy. (**A**–**C**) CEP suppresses melanoma tumor growth in vivo. C57BL/6 mice (n = 5 per group) bearing subcutaneous B16 melanoma tumors (2 × 10^5^ cells) were treated with vehicle or CEP (20 or 40 mg/kg, intraperitoneally). (**A**) Representative tumor images at endpoint. (**B**) Tumor growth curves monitored daily. (**C**) Tumor weights measured at endpoint. (**D**) CEP increases intratumoral infiltration of CD8^+^ T cells. Representative immunofluorescence staining of CD8^+^ T cells in tumor sections, with bar graphs quantifying CD8^+^ T cell percentages. Scale bar: 5 μm. (**E**) Schematic diagram of the experimental design for CD8^+^ T cell depletion and anti–PD-1 combination therapy studies. (**F**–**H**) Depletion of CD8^+^ T cells abrogates the antitumor effect of CEP. Mice were grouped as: control, anti-CD8 (αCD8; 200 μg/mouse, intraperitoneally every 72 h), CEP (40 mg/kg daily), or αCD8 + CEP. (**F**) Representative tumor images. (**G**) Tumor growth curves. (**H**) Tumor weight analysis at endpoint. (**I**–**K**) CEP synergizes with anti–PD-1 therapy to inhibit tumor growth. Mice were treated with control, anti–PD-1 antibody (αPD-1; 200 μg/mouse, intraperitoneally every 72 h), CEP (40 mg/kg daily), or αPD-1 + CEP. (**I**) Representative tumor images. (**J**) Tumor growth curves. (**K**) Tumor weight analysis. Data represent mean ± SEM from at least three independent experiments. ** *p* < 0.01 and *** *p* < 0.001 indicate statistical significance. CEP, cepharanthine; αCD8, anti-CD8 antibody; αPD-1, anti-PD-1 antibody.

## Data Availability

The original contributions presented in the study are included in the article/Appendix A. Data will be made available on request.

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
