# Peer review of "Cepharanthine Enhances MHC-I Antigen Presentation and Anti-Tumor Immunity in Melanoma via Autophagy Inhibition"

_cells, 2025, doi:10.3390/cells14161231_

Round 1

Reviewer 1 Report

Comments and Suggestions for Authors

The authors present a comprehensive study investigating the immunomodulatory effects of Cepharanthine (CEP) in melanoma models. The work identifies CEP as a late-stage autophagy inhibitor that enhances MHC-I-mediated antigen presentation by disrupting lysosomal acidification, thereby promoting CD8+ T cell-mediated anti-tumor immunity. While some limitations exist, the overall quality and significance justify publication after addressing the major points raised.

Major Comments:

Introduction:

  1. The Introduction lists the properties and actions of CEP but does not compare it to other known autophagy inhibitors or potential therapies modulating MHC-I. The authors could highlight what distinguishes CEP and emphasise the uniqueness of their approach.
  2. The Introduction describes the study results (lines 76-82), which is more appropriate for an abstract or conclusion rather than an introduction. The Introduction should end with a clear statement of the study’s aims and hypotheses, not its findings.

Flow Cytometry Analysis (section 2.4)

Although the flow cytometry methods follow standard procedures, more detail and clarity are needed regarding marker selection, gating strategies, controls, fluorescence compensation, and sample handling.

  1. The method mentions adding "appropriate fluorochrome-conjugated antibody" but does not specify which markers were used to define cell populations, which limits assessment of specificity and reproducibility.
  2. The flow cytometry methods lack detailed gating strategies. The flow cytometry methods refer simply to "appropriate gating strategies" without detailed descriptions.
  3. Trypsin treatment may degrade or alter surface epitopes, potentially biasing results. No mention is made of steps to minimise this effect.
  4. While mean fluorescence intensity (MFI) measurements are mentioned, there is no explanation of how MFI data were normalised or interpreted relative to protein expression. Without clear normalisation criteria (e.g., relative to negative controls), interpretation of MFI values remains unclear.
  5. Tumor cell lines often exhibit autofluorescence, which may interfere with fluorochrome signals and should be accounted for.

In Vitro Cytotoxicity Assay (section 2.5):

  1. The ratio of CD8+ T cells to tumor cells and the co-culture duration are not specified.
  2. It is unclear whether untreated controls or T cell activity inhibitors were included to validate the assay and interpret changes in cytotoxicity properly.

GFP-LC3 Autophagosome Detection (section 2.7):

  1. Counting puncta manually is susceptible to observer bias and requires standardised criteria or, preferably, automated image analysis to ensure consistency.

Immunofluorescence Detection of Autophagosome-Lysosome Fusion (Section 2.9):

  1. Imaging only 8-10 randomly selected fields may be insufficient for statistical representativeness.
  2. There is no indication of fluorescence intensity normalization or inclusion of negative controls to ensure specificity.

Immunohistochemical Staining (section 2.11)

  1. Insufficient detail on quantification and image analysis: The methodology lacks descriptions of quantitative criteria or software-based analysis for CD8+ T cell infiltration.
  2. Specificity controls for antibodies are not mentioned.

Animal Tumor Model (section 2.12)

  1. The authors provide limited data on CEP’s absorption, distribution, metabolism, or systemic toxicity. Additionally, dose justification is lacking, leaving uncertainty about potential toxicity.
  2. The protocol does not specify how long after tumor cell injection treatments were initiated.
  3. Small number of animal groups (n=5)

Discussion:

1. The discussion mentions several limitations (e.g., lack of identification of a specific molecular target in the V-ATPase complex and the need for future research). However, it would benefit from a more detailed discussion of potential barriers to clinical translation, such as differences between animal models and humans, possible toxicities, and limitations of the tumor model.

Reviewer 2 Report

Comments and Suggestions for Authors

This original study investigates the immunomodulatory role of cepharanthine (CEP), a natural bisbenzylisoquinoline alkaloid, in enhancing MHC-I-mediated antigen presentation and anti-tumor immunity in melanoma. CEP significantly upregulates both surface and intracellular MHC-I expression in melanoma cells by inhibiting autophagic degradation, leading to improved antigen presentation and increased susceptibility to CD8⁺ T cell-mediated cytotoxicity. Mechanistically, CEP disrupts lysosomal acidification—a critical step in late-stage autophagy—without affecting autophagosome–lysosome fusion. This inhibition blocks the degradation of MHC-I complexes and promotes their accumulation. The in vivo results show that CEP slows down tumor growth by relying on CD8⁺ T cells and works well with anti–PD-1 therapy, leading to better tumor shrinkage and more CD8⁺ T cells getting into the tumor. These results suggest that CEP could be a valuable addition to cancer treatments, providing a new way to tackle how tumors avoid the immune system by focusing on the autophagy–MHC-I pathway.

This study is carefully planned and clearly presented, showing how CEP improves the display of antigens by specifically blocking late-stage autophagy. It includes comprehensive in vitro and in vivo experiments with appropriate controls and statistical rigor. Targeting lysosomal acidification to modulate MHC-I is a relatively unexplored strategy with strong translational potential. Demonstrating enhanced efficacy with anti–PD-1 therapy increases clinical relevance. The minimal toxicity and non-cytotoxic concentrations of CEP support its potential for combination regimens.

However, some issues need further improvement.

The precise molecular component of the lysosomal V-ATPase complex affected by CEP remains unknown. This limits mechanistic depth, which must be discussed.

Without data on systemic exposure or tumor penetration, clinical translation is premature. This must also be further discussed.

The study focuses solely on melanoma; generalizability to other MHC-I–deficient tumors is speculative.

The B16 and A375 models may not fully recapitulate the heterogeneity of human melanoma. This must be discussed as well.

While the short-term toxicity is addressed, the chronic effects of autophagy inhibition have not been explored. Why?

According to this, a major revision is required. 

Round 2

Reviewer 1 Report

Comments and Suggestions for Authors

The authors have addressed the majority of the comments, and the work in its current form is acceptable.

Reviewer 2 Report

Comments and Suggestions for Authors

Thanx for the authors for accepting the suggestions.

The revised version is now acceptable for publication.